# Circadian Rhythm and Nitrogen Metabolism Participate in the Response of Boron Deficiency in the Root of *Brassica napus*

**DOI:** 10.3390/ijms25158319

**Published:** 2024-07-30

**Authors:** Ling Liu, Xianjie Duan, Haoran Xu, Peiyu Zhao, Lei Shi, Fangsen Xu, Sheliang Wang

**Affiliations:** 1National Key Laboratory of Crop Genetic Improvement, Huazhong Agricultural University, Wuhan 430070, China; hzaulingliu@webmail.hzau.edu.cn (L.L.); leish@mail.hzau.edu.cn (L.S.); fangsenxu@mail.hzau.edu.cn (F.X.); 2Microelement Research Centre, Huazhong Agricultural University, Wuhan 430070, China; xhr15093322403@163.com (H.X.); zhaopeiyu618618@163.com (P.Z.); 3College of Resources and Environment, Huazhong Agricultural University, Wuhan 430070, China; xianjieduan@mail.hzau.edu.cn

**Keywords:** boron deficiency, root development, circadian rhythm, nitrogen metabolism

## Abstract

Boron (B) deficiency has been shown to inhibit root cell growth and division. However, the precise mechanism underlying B deficiency-mediated root tip growth inhibition remains unclear. In this study, we investigated the role of *BnaA3.NIP5;1*, a gene encoding a boric acid channel, in *Brassica napus* (*B. napus*). *BnaA3.NIP5;1* is expressed in the lateral root cap and contributes to B acquisition in the root tip. Downregulation of *BnaA3.NIP5;1* enhances B sensitivity in *B. napus*, resulting in reduced shoot biomass and impaired root tip development. Transcriptome analysis was conducted on root tips from wild-type *B. napus* (QY10) and *BnaA3.NIP5;1* RNAi lines to assess the significance of B dynamics in meristematic cells during seedling growth. Differentially expressed genes (DEGs) were significantly enriched in plant circadian rhythm and nitrogen (N) metabolism pathways. Notably, the circadian-rhythm-related gene *HY5* exhibited a similar B regulation pattern in *Arabidopsis* to that observed in *B. napus*. Furthermore, *Arabidopsis* mutants with disrupted circadian rhythm (*hy5*/*cor27*/*toc1*) displayed heightened sensitivity to low B compared to the wild type (Col-0). Consistent with expectations, B deficiency significantly disrupted N metabolism in *B. napus* roots, affecting nitrogen concentration, nitrate reductase enzyme activity, and glutamine synthesis. Interestingly, this disruption was exacerbated in *BnaA3NIP5;1 RNAi* lines. Overall, our findings highlight the critical role of B dynamics in root tip cells, impacting circadian rhythm and N metabolism, ultimately leading to retarded growth. This study provides novel insights into B regulation in root tip development and overall root growth in *B. napus*.

## 1. Introduction

Boron (B) is an essential micronutrient for higher plants, while the suitable range of B concentration that plants can absorb is very narrow. In soil, B mainly exists in the form of boric acid (H_3_BO_3_), which is easily leached in areas with high rainfall. Consequently, in countries such as China, Brazil, and Northern Europe, where precipitation is abundant, available B in the soil is usually low [1]. According to reports, over 100 crops in these countries/regions are suffering from B deficiency [2].

B plays an important role in plant growth and development, mainly including the structure and function of cell walls, membrane integrity, and the transportation and distribution of carbohydrates [3,4]. B deficiency has a negative impact on the physiology and morphology of various stages of vascular plant development [5,6]. Some physiological processes such as hormone synthesis, sugar transport, protein metabolism, amino acid synthesis, carbohydrate metabolism, RNA synthesis, pollen tube formation, nucleotide synthesis, and phenol metabolism in plants are seriously affects by B deficiency [6]. B is essential for actively growing regions of plants, such as root tips, new leaves, and bud development. In the aboveground part, due to the limited mobility of B in the phloem, meristematic tissue struggle to obtain sufficient B to sustain growth under low-B conditions [7]. Low B significantly inhibits the growth and development of plant roots. B deficiency inhibits the growth of primary and lateral roots but elicits root hair elongation and formation [8]. It was depicted by Martín-Rejano et al. [9] that B deprivation alters the cell elongation within the plants rather than inhibiting the cell division of meristematic tissues. The impediment of cell elongation process is directly related to the adversity caused by B deficiency to the physical integrity of cell walls that further leads to abnormality in cell wall growth [10]. However, a series of evidence suggests that B deficiency severely inhibits the activity of *CYCB1;1* in root tips, reduces the number of meristem cells, and shortens the length of meristem regions [8,11].

Under low-B conditions, plants actively absorb B from the soil through channel and transport proteins. In *Arabidopsis*, *AtNIP5;1* is mainly expressed on the cell membrane of the lateral root cap and epidermal cells and is responsible for B uptake in root [12,13]. In the non-hair zone of the root, where no vascular tissues are established, the expression of *AtNIP5;1* in these cells was proposed to contribute to B uptake for root tip growth. In *Brassica napus*, *BnaA3.NIP5;1* is expressed specifically on the plasma membrane of in lateral root cap, and the *RNAi* operation largely reduced B accumulation in *B. napus* root tip, leading to retarded root growth [14]. From the anatomical structure of the roots, the lateral root cap surrounds the meristematic zone. Therefore, the meristem zone should be the starting point of *BnaA3.NIP5;1*-mediated root tip growth. The functional deficiency of *BnaC4.BOR2* inhibited primary root length, significantly reducing B concentration in the roots of *B. napus* under low-B stress, and *BnaC4.BOR2* is localized in lateral root cap, too [15]. The stem cell niche and the root meristem size in plants are influenced by intercellular interactions and signaling networks [16]. External environmental factors, such as light–dark cycles, impact the global transcriptome oscillation in roots [17]. Interestingly, single-cell oscillations in the root tip exhibit higher amplitudes and expression levels compared to other sections of the plant [18]. Additionally, nutrient availability, particularly nitrogen, affects root growth [19]. The specific form of nitrogen even regulates the elongation and division of root tip cells [20]. As a dynamic region of gene oscillation, the root tip cells under low-B conditions possess important pathways related to gene response and physiological processes. Beyond its role as a pectin glue for Rhamnogalacturonan-II dimer formation, we are keenly interested in understanding how external signals and intracellular physiological processes converge in B-deficient root tips.

Allotetraploid rapeseed (*Brassica napus* L., AnAnCnCn, 2n = 38), a major oil crop species globally, exhibits a high demand for B and is exceptionally sensitive to B deficiency [21,22]. The specific expression of *BnaA3.NIP5;1* in the lateral root cap, along with the establishment of *BnaA3.NIP5;1* RNAi lines (NSQ *B. napus*) [14], positions *B. napus* as an ideal material for exploring novel insights into B-related root tip growth. In this study, we employed RNA-seq technology to investigate how B deficiency impacts root development in *B. napus* root tips. By utilizing the *BnaA3.NIP5;1* RNAi lines to create lower B conditions, our aim was to reveal key physiological pathways and integrate measurements of physiological indicators, providing novel insights into root development mechanisms under low-B conditions.

## 2. Results

### 2.1. Downregulation of BnaA3.NIP5;1 Deforms B. napus Root Architecture under Boron-Deficient Conditions

RNA interference (RNAi) targeting *BnaA3.NIP5;1* resulted in its reduced expression in the root tips of *B. napus* compared to the wild type ‘QY10’ [14]. This downregulation led to diminished B accumulation in the root tips relative to ‘QY10’. Consequently, the suppressed expression of *BnaA3.NIP5;1* impeded seedling development under low-B conditions [14]. To elucidate the impact of *BnaA3.NIP5;1* downregulation in the meristem zone on seedling growth under B stress, phenotypic comparisons were conducted between NSQ lines (*BnaA3.NIP5;1* RNAi) and ‘QY10’, both cultivated in plate and hydroponic systems. Under standard B conditions, no growth discrepancies were observed between ‘QY10’ and NSQ seedlings grown on plates (Appendix A). However, under low-B stress, the NSQ lines exhibited significantly restricted primary root elongation and lateral root formation compared to ‘QY10’ (Figure 1A,B). Notably, the cellular arrangement at the root tips was markedly disorganized in NSQ lines (Appendix A and Figure 1C–F). Consistent with observations in plate, the NSQ lines displayed stunted shoot and root growth relative to ‘QY10’ in a hydroponic system under low-B conditions (Figure 1G and Appendix A). Additionally, the non-root hair zone length was shorter in NSQ lines than in ‘QY10’ when subjected to low-B stress (Figure 1H–K). These findings indicate that the downregulation of *BnaA3.NIP5;1* adversely affects the root architecture of *B. napus* under B-deficient conditions.

### 2.2. Transcriptome Analysis 

To delve deeper into the genome-wide dynamics of gene expression, we conducted RNA-seq analysis on 0.5 cm root tips (100 μM B treated ‘QY10’, 0.25 μM B treated ‘QY10’ and NSQ lines). The sample RNA has high integrity and can be used for subsequent experiments (Appendix A). The alignment of clean reads to the *B. napus* reference genome yielded a match rate between 93.91% and 94.2% (Appendix A). Both sample correlation and principal component analysis (PCA) confirmed high reproducibility among samples, substantiating the reliability of our data for subsequent analysis (Figure 2A,B). Transcriptomic exploration identified a total of 50,493 genes, with 44,749 genes exhibiting consistent coexpression across all samples, suggesting a foundational set of active genes pertinent to this study (Figure 2C). 

To pinpoint the genes influencing root tip development in B-deficient conditions, we analyzed differentially expressed genes (DEGs) across various comparisons. The comparison between 0.25 μM B treated ‘QY10’ (LB_QY10) and 100 μM B treated ‘QY10’ (NB_QY10) revealed 1215 DEGs (560 up and 655 down), while the contrast between LB-NSQ and NB_QY10 manifested 4982 DEGs (2724 up and 2258 down) (Figure 2D). This suggests an escalation in gene involvement in response to B stress at the NSQ root tip as B levels diminish. Additionally, the comparison between LB_NSQ and LB_QY10 uncovered 2409 DEGs (1591 up and 819 down) (Figure 2D), indicating these genes’ specific responsiveness to acute B deficiency at the root tips. Further analysis indicated the presence of 126 DEGs across all groups (Figure 2E). The clustering of these 126 DEGs revealed that 74 were upregulated and 34 downregulated, signifying a predominantly consistent genetic response to B stress (Appendix A and Figure 2F). To corroborate the RNA-seq findings, we selected 5 upregulated and 5 downregulated DEGs at random for qRT-PCR validation. The concordance between the qRT-PCR results and the RNA-seq data reaffirms the veracity of our RNA-seq analysis (Figure 3 and Appendix A).

### 2.3. GO Functions and KEGG Pathway Analysis of DEGs 

The Gene Ontology (GO) analysis of DEGs across various groups revealed categorization into three domains: biological processes (BP), cellular components (CC), and molecular functions (MF). In the LB_QY10 versus NB_QY10 comparison, GO terms were significantly enriched with 238 in BP, 23 in CC, and 230 in MF (Appendix A). Notably, processes such as metal ion transport, amino polysaccharide decomposition metabolism, chitin metabolism, amino sugar metabolic, and cell wall macromolecule catabolic process were markedly prevalent in BP (Appendix A)**.** The CC category was predominantly characterized by the extracellular region, including the cell wall (Appendix A). Key MFs included activities related to metal ion transmembrane transporters, inorganic ion transmembrane transporter, cation transmembrane transporter, and chitin binding (Appendix A). In the LB_NSQ versus LB_QY10 group, there were 428 BP, 91 CC, and 314 MF GO terms enriched (Appendix A). In addition to the metal ion transport, this group highlighted additional processes like nicotianamine-related pathways, tricarboxylic acid biosynthesis, and response to stimulus in BP (Appendix A). The CC was largely represented by the extracellular region and apoplast (Appendix A). MF enrichment was significant for transporters and enzymes associated with metal ions and the antioxidant system (Appendix A). 

In the LB_NSQ versus NB_QY10 comparison, a reduced number of GO terms was observed in the CC category (Appendix A). BP enrichment included responses to chemicals and transport of drugs and inorganic anions (Appendix A), while MF was characterized by activities such as antiporter, drug transmembrane transporter, and O-acetyltransferase (Appendix A). The DEGs in the LB_NSQ vs. LB_QY10 vs. NB_QY10 group delineate the fundamental gene response to low-B stress in root tips, with 76 BP, 4 CC, and 52 MF enriched GO terms (Appendix A) These terms were significantly associated with polyamine biosynthesis and metabolism in BP, the extracellular region in CC, and activities such as metal ion transmembrane transport in MF (Appendix A). 

Subsequent KEGG analysis identified metabolic pathways impacted by these DEGs. For LB_QY10 versus NB_QY10, enriched pathways included circadian rhythm-plant and various metabolic processes like glutathione and cyanoamino acid metabolism (Figure 4A). In LB_NSQ versus NB_QY10, pathways such as nitrogen metabolism, glutathione metabolism, glycolysis/gluconeogenesis, circadian rhythm-plant, and flavonoid biosynthesis were prominent (Figure 4B). Comparing LB-NSQ versus LB_QY10 highlighted pathways like nitrogen metabolism, circadian rhythm-plant, alanine, aspartate, and glutamate metabolism (Figure 4C). Notably, circadian rhythm-plant and nitrogen metabolism pathways were consistently altered across groups (Figure 4A–F). This suggests that circadian rhythm regulation and nitrogen metabolism are key to the root tip’s response to B deficiency in *B. napus*.

### 2.4. Circadian Rhythm-Plant Genes Are Responsive to Boron Deficiency in Root Tips

To elucidate the circadian rhythm-plant pathway’s response to B deficiency, we analyzed the network of this pathway (Appendix A). Genes such as *LHY*, *HYH*, *ELF3*, *HY5*, *SPA1*, and *TT4* (highlighted in red) exhibited increased expression, correlating with reduced B levels in root tips (Figure 5A and Appendix A), with a 1- to 41-fold increase observed in LB_QY10 vs. NB_QY10. This upregulation was further enhanced in LB-NSQ vs. NB_QY10 (Figure 5A and Appendix A), indicating a progressive and strengthening response to diminishing B concentration. Notably, *LHY* genes (*BnaA10G0008600ZS* and *BnaC05G0010100ZS*) showed the highest fold increase despite low basal expression levels (Figure 5A and Appendix A). *HY5* and *TOC1* (*BnaC07G0381000ZS*) genes had the highest non-induced expression levels; however, only *HY5* demonstrated significant upregulation under low-B conditions (Figure 5A and Appendix A). Conversely, *PRR3* and *GI* expression decreased with lower B levels (highlighted in green in Figure 5A, Appendix A). The response of other genes such as *PRR5s*/*7s*/*9s*, *TOC1s*, and *COR27*/*28s* to varying B levels was inconsistent (highlighted in purple in Figure 5A and Appendix A). 

Interestingly, the *AtHY5* gene exhibited a 2.2-fold upregulation by low-B treatment in roots (Figure 5B). This response pattern is similar to that observed for BnaA10G0237000ZS (*HY5*) in *B. napus* roots, despite the differences in B levels (Figure 5B and Appendix A). These findings suggest that *HY5* maintains a stable response within a specific B concentration range. Additionally, β-Glucuronidase (GUS) reporter-assisted histochemical analysis indicated that low-B conditions induced *AtHY5* expression in roots, hypocotyls, cotyledons, and leaves (Figure 5C–L). Collectively, these findings demonstrate that circadian rhythm-plant genes are responsive to B deficiency in root tips.

### 2.5. Expression of Circadian-Rhythm-Related Genes Alters Low-B Sensitivity in Arabidopsis Roots

To investigate the impact of circadian rhythm on root growth and development under B deficiency, we compared the phenotypes of *lhy*, *hy5*, *cor27*, and *toc1* mutants with Col-0 at different B levels in *Arabidopsis*. Notably, no visible differences were observed between *lhy* mutants and Col-0, regardless of B levels. Under normal B conditions, the functional deficiency of *HY5* inhibited primary root growth by 30% ± 10% compared to Col-0, while under low-B conditions, this inhibition increased to 58% ± 6%. Additionally, the functional deficiency of *COR27* resulted in a longer primary root length (increased by 15% ± 5% compared to the wild type) under normal B conditions but a 24% ± 6% reduction under low-B conditions compared to Col-0. Interestingly, the functional deficiency of *TOC1* did not exhibit any significant root length difference compared to Col-0 under normal B conditions, but it showed a 28% ± 3% reduction under low-B conditions (Figure 6A,C). Further analysis of the root tip structure revealed that the expression of these genes had varying impacts on division and elongation in response to B deficiency. Loss of *HY5* function consistently led to retarded division, elongation, and lateral root formation, regardless of B levels (Figure 6B,C). Moreover, the *cor27* mutant and *toc1* mutants exhibited a shorter elongation zone than Col-0 under low-B conditions, and *toc1* mutants displayed reduced lateral root development compared to Col-0 under the same conditions. These findings highlight the role of circadian rhythm pathway genes in altering *Arabidopsis* root sensitivity to low B levels.

### 2.6. B Deficiency Impairs Nitrogen Uptake and Assimilation in B. napus

In the LB_QY versus NB_QY comparison, all nitrate transporter genes *NRT2;1* exhibited minimal downregulation (1.1~1.7 fold), while in the LB-NSQ versus NB_QY comparison, thy were significantly downregulated (4.2–41.1 fold) (Figure 7A and Appendix A). Notably, *NRT2;3*, *NRT2;5*, and *NRT2;6s* showed even greater downregulation in the LB_QY versus NB_QY comparison and were further reduced in the LB-NSQ versus NB_QY comparison (Figure 7A and Appendix A). These findings suggest that B deficiency leads to an overall decrease in the expression levels of nitrate transporters in root tips, potentially affecting nitrate uptake. Subsequently, we assessed the nitrogen concentrations in the roots of *B. napus* seedlings subjected to normal and low-B conditions. While N concentrations were comparable between QY10 and NSQ lines under normal B treatment, B–deprived roots accumulated more N, with NSQ lines exhibiting higher N concentrations than QY10 roots (Figure 7C).

We then focused on DEGs related to nitrogen assimilation (Figure 7B). *NIA1s* (nitrate reductase, NR) were downregulated by 2.2–2.6 fold in the LB_QY versus NB_QY comparison and 4.3–5.3 fold in the LB-NSQ versus NB_QY comparison (Figure 7A and Appendix A). Similarly, *GLN1;2* and *GLN1;4* (glutamine synthase gene, GS) were downregulated by 2.6-fold and 2.2-fold, respectively, in the LB-NSQ versus NB_QY comparison (Figure 7A and Appendix A). Consistent with these observations, the activities of NR and GS were largely inhibited in *B. napus* roots under low-B conditions (Figure 7D,E). Notably, NSQ lines exhibited lower NR and GS activities than QY10 lines. In contrast, the glutamine dehydrogenase (GDH) showed higher enzyme activity under normal B treatment, aligning with its gene regulatory response (Figure 7A,F and Appendix A). These results highlight how B deficiency adversely affects N uptake and assimilation in *B. napus* roots. 

## 3. Discussion

The root tip of a plant consists of two main zones: the meristem zone (MZ) and the elongation zone (EZ). When there is a deficiency of B, it significantly restricts root tip growth and development. This deficiency affects the cell activity in the MZ and can even lead to the absence of EZ cells [8,11]. Loss of function in the *AtNIP5;1* gene result in severe root growth inhibition under low-B conditions [12]. *AtNIP5;1* is responsible for B supply in root tips, and its impairment affects B uptake [12]. Similar findings were observed in *BnaA3.NIP5;1* RNAi lines (NSQ lines) under low-B conditions [14]. Despite *BnaA3.NIP5;1* being specifically expressed on the lateral root cap cells surrounding the MZ [10], both MZ and EZ showed significant retarded growth compared to wild-type QY10 lines under low-B conditions (Figure 1). This retarded growth may be due to the decrease in cell division caused by B deficiency, which affects the supply of transition cells for elongation. Additionally, EZ cells have high B requirements [23], and the lower B levels in NSQ root tips relative to QY10 may also impact cell wall integrity, considering B contribution to RG-II dimer formation [24]. 

The EZ and MZ play crucial roles in gene regulation within plant root tips. However, the transcriptional response to B deprivation in *B. napus* root tips remains poorly understood. To shed light on this, researchers analyzed mRNA sequencing data from B-deprived *Arabidopsis* roots [23]. They discovered that genes related to various responses (such as chitin, wounding, ROS, jasmonic acid, and abscisic acid), oxidation–reduction processes, cell wall organization, and flavonoid glucuronidation were differentially regulated. These findings align with our previous transcriptomic analyses [25] and other studies [26] in *Arabidopsis*. In *B. napus* root tips (5 mm), differentially expressed genes (DEGs) in the B-efficient genotype ‘QY10’ responded to B deficiency by enriching processes related to ion transport, chitin catabolism, metabolic processes, cell wall functions, and amino sugar metabolism (Appendix A). Further B reduction in the root tips (in the B-inefficient genotype ‘NSQ’) disrupted nicotianamine metabolism, tricarboxylic acid (TCA) biosynthesis, oxidoreductase activity, etc. (Appendix A). Notably, ‘QY10’ being B-sufficient, exhibited higher expression of *BnaA3.NIP5;1* in root tips compared to the B-insufficient cultivar ‘W10’ resulting in stronger B homeostasis under low-B conditions [14]. In essence, extreme B levels caused by downregulation of *BnaA3.NIP5;1* disrupted nicotinamine metabolism and TCA biosynthesis in *B. napus* root tips. Interestingly, low-B-treated *Arabidopsis* root tips (8 mm) showed upregulation of genes encoding nicotinamine synthase, glycine-rich protein, hydroxyproline-rich protein, branched-chain amino acid aminotransferase, and phosphoethanolamine N-methyltransferase [27]. This suggests that amino acid and nitrogen assimilation processes play a crucial role in the physiological response to B deficiency in root tips.

Nitrate levels rise as a consequence of reduced nitrate reductase activity in various B-deficient plants [28]. Nitrogen metabolism pathway was highlighted in all KEGG comparisons (Figure 4A–C), suggesting that N metabolism is a characteristic response of B deficiency. Consequently, B deficiency increased N levels and downregulated NRTs expression, NR activity, and GS activity (Figure 7A–E), which were further up- or downregulated, respectively. Glutamate dehydrogenase (GDH) is a key enzyme involved in both carbon and nitrogen metabolism. It catalyzes the reversible oxidative deamination of glutamate into α-ketoglutarate and ammonia [29]. GDH activity exhibited an increase in QY10 roots and was further enhanced in NSQ lines (Figure 7F). Notably, either the downregulation of NRTs, NR, and GS activities or the upregulation of GDH activity signifies disturbances in nitrogen assimilation and metabolic processes (Figure 8). B deficiency has a straightforward impact on cell growth, potentially leading to an excess of nutrients such as N, P, and K relative to cellular requirements.

Research indicates that the interaction between B and N significantly influences the absorption and utilization of other nutrients (including P, K, Ca, and Mg), ultimately affecting plant growth and productivity [30,31]. In *B. napus*, B deficiency accelerates leaf aging due to induced N starvation stress, disrupting the balance of C and N pools [32]. Conversely, B fertilizer application improves N absorption, utilization efficiency, and seed yield in *B. napus* [33]. These findings underscore the importance of root B supply for nitrogen assimilation in *B. napus*.

The 24 h rhythm, driven by Earth’s rotation, serves as a fundamental time-generating mechanism. In plants, the circadian clock is believed to be an evolutionary adaptation that maintains growth homeostasis and mitigates various environmental stresses [34]. Extensive research has demonstrated that the circadian clock perceives external signals and orchestrates 24 h rhythms and periodic oscillations in synchrony with daily life, coordinating physiological and metabolic processes [35]. For instance, the circadian rhythm regulates the availability of light by modulating stomatal opening and closing, which impacts leaf photosynthesis [36,37]. One component of the circadian clock, *LHY* (late elongated hypocotyl), is associated with circadian oscillations in light/dark cycles, influencing light-harvesting and chlorophyll synthesis [38,39]. Surprisingly, the expression of LHYs in root tips under normal B conditions is very low (Appendix A) suggesting that they may not play a significant role in root growth control. Indeed, despite the high upregulation induced by low B levels, we did not observe growth differences between Col-0 and *Atlhy* mutants under low-B conditions.

In contrast to the *LHY* genes, *HY5* (ELONGATED HYPOCOTYL5) and *TOC1* (TIMING OF CAB EXPRESSION1) exhibited the highest expression levels under normal B conditions (Appendix A), suggesting their potential roles in growth regulation. As anticipated, Arabidopsis mutants lacking *HY5* or *TOC1* displayed stunted root growth compared to the wild-type Col-0 under low-B conditions (Figure 6). Notably, the *hy5* mutant also exhibited reduced primary root length even under normal B conditions (Figure 6), emphasizing HY5’s constitutive involvement in mediating root growth. *HY5* encodes a critical transcription factor downstream of photoreceptors, governing plant photomorphogenesis in both shoot and root tissues [40,41,42,43,44]. Recent work by Li et al. (2024) demonstrated that *HY5* plays a distinct role in root development by regulating reactive oxygen species (ROS) balance in the root apical meristem, independent of light perception in aboveground organs [45]. Interestingly, while light-triggered Fenton chemistry induces root growth inhibition under phosphate deficiency [46], light does not significantly impact root growth under B deficiency [47]. The pronounced growth inhibition observed in the *hy5* mutant under low-B conditions may be attributed to ROS imbalance, as *HY5* directly activates *PER6* expression to eliminate hydrogen peroxide [45].

The plant circadian oscillator relies on essential components, including *TOC1* (PRR1) [48,49]. Recent research highlights the significance of proper *TOC1* expression in maintaining cellular energy levels and orchestrating diel and circadian oscillations of sugars, amino acids, and tricarboxylic acid (TCA) cycle intermediates [50]. Notably, reducing B levels in NSQ root tips disrupted the TCA process (Appendix A) and *Arabidopsis toc1* mutants exhibit shorter primary roots than Col-0 under low-B conditions (Figure 6). The TCA cycle, intricately linked to amino acid synthesis and metabolism, likely connects nitrogen assimilation with the circadian clock through key genes like *TOC1*. These two pathways were simplified and integrated into a working model (Figure 8).

In *Arabidopsis*, the cold-responsive genes COLD-REGULATED GENE27 (*COR27*) and *COR28* are regulated by the circadian clock [51,52]. These genes repress the transcriptional expression of *TOC1* and *PRR5/7/9* through direct chromatin binding, integrating circadian clock, cold stress, and flowering time [53,54]. Intriguingly, in *B. napus* root tips, B deficiency downregulates the expression of *COR27*, *COR28*, *TOC1*, and *PRRs* genes (Figure 5A and Appendix A), suggesting that *COR27* and *COR28* may not directly regulate *TOC1* and *PPRs*. Notably, the *cor27 Arabidopsis* mutant exhibited longer primary roots under normal B conditions and shorter primary roots under normal low-B conditions compared to Col-0 (Figure 6). However, the precise mechanism by which *COR27* and *COR28* participate in the plant’s low-B response remains unknown.

## 4. Materials and Methods 

### 4.1. Plant Materials and Cultivation

The wild-type *B. napus* ‘QY10’ and the RNAi lines ‘NSQ’ (*BnaA3.NIP5;1*) utilized in this study were developed by He et al. (2022) and are preserved in our laboratory [14]. For hydroponic cultivation, robust seeds were surface-sterilized with 1% NaClO for 15 min and thoroughly rinsed with ultrapure water (18.25 MΩ·cm). Post-vernalization at 4 °C for one day, the sterilized seeds were sown on damp gauze. Following a 5-day germination period, uniform seedlings were transferred to 10 L black plastic containers containing Hoagland’s solution, supplemented with either 0.25 or 100 μM B, and grown for 15 days. Growth parameters were systematically recorded. 

For RNA extraction, all samples were initially cultured hydroponically with standard B concentration (100 μM) for three weeks. Subsequently, roots were washed thrice with ultrapure water and then exposed to 0.01 and 100 μM B for 24 h. A 0.5 cm segment from the root tip was sampled, instantly frozen in liquid nitrogen, and stored at −80 °C. All experimental procedures were conducted in triplicate, with nutrient solutions refreshed every five days. Plant cultivation occurred in a controlled growth room at 24 °C, under an 8/16 h light/dark cycle, and a photon flux density of 300–320 μmol m^−2^ s^−1^.

Arabidopsis seeds were surface-sterilized for 10 min using 1% NaClO after a 1 min sterilization with 75% ethanol, followed by thorough rinsing with sterile ultrapure water (18.25 MΩ·cm). The seeds were sown on solid MGRL medium (including 1.75 mM sodiumphosphate buffer (pH 5.8), 1.5 mM MgSO_4_, 2.0 mM Ca(NO_3_)_2_, 3.0 mM KNO_3_, 67 µM Na_2_EDTA, 8.6 µM FeSO_4_, 10.3 µM MnSO_4_, 30 µM H_3_BO_3_, 1.0 µM ZnSO_4_, 24 nM (NH_4_)_6_Mo_7_O_24_, 130 nM CoC1_2_, 1 µM CuSO_4_) with the indicated B concentration after a 2-day vernalization at 4 °C. The plates were then vertically positioned in a growth chamber set to a 16 h light/8 h dark cycle at 22 °C, where they were grown for 12 days. Growth parameters were measured.

### 4.2. Characterization of Root Growth

Root tips of *B. napus* (0.5 cm) were harvested after 10 days of culture on agar plates with 0.1 μM B and immediately fixed in plant tissue fixative. The sampled were embedded in paraffin, and then sectioned and stained with Safranin O-Fast Green, and finally imaged under an inverted microscope (DS-Ri2, Nikon, Tokyo, Japan). Additionally, *B. napus* roots were harvested after 15 days of hydroponic culture with either 0.25 or 100 μM B and examined under the bright field of a Nikon DS-Ri2 optical microscope. The microstructure of *Arabidopsis* roots was visualized using a laser confocal microscope (TCS-SP8; Leica, Wetzlar, Germany) with a maximum excitation/emission wavelength of 535/617 nm. This was performed after a 30 min staining period with propidium iodide (PI) to ensure clear imaging of the root architecture.

### 4.3. RNA-seq Data Analysis

Total RNA was extracted using TRIZOL Reagent (Vazyme, Nanjing, China) following the manufacturer’s protocol. RNA concentration, purity, and integrity were assessed using a NanoDrop 2000 spectrophotometer (Thermo Fisher, Waltham, MA, USA) and an Agilent 5400 bioanalyzer (Agilent Technologies, Santa Clara, CA, USA). High-quality RNA samples were selected for sequencing on the Illumina NovaSeq 6000 platform, with library construction performed at Novogene Corp (Beijing, China).

To ensure data quality and reliability, adapter sequences and low-quality reads (base number Qphred ≤20 accounting for more than 50% of the read length or containing unknown bases ‘N’) were filtered out using fastp software (version 0.19.7) [55]. HISAT2 software (version 2.0.5) was employed to align Clean Reads with the reference genome swiftly and accurately, obtaining read localization information [56]. The reference genome *Bra_napus_v2.0* was sourced from the National Center for Biotechnology Information database (https:www.ncbi.nlm.nih.govdatasetsgenomeGCF_000686985.2 accessed on 28 July 2024) (version 2022-9-13). New transcripts were assembled using StringTie software (version 1.3.3b) [57].

### 4.4. Identification of Differential Expression Gene 

The expression abundance and variations of the transcripts were quantified by calculating the FPKM (fragments per kilobase of transcript per million fragments mapped) values using feature counts software. Transcripts with an FPKM value of ≥1 were considered actively expressed genes [58,59]. Differential expression gene (DEGs) among the samples were identified by the criterion (|log2Fold Change| ≥ 1 and padj ≤ 0.05), performed by software DESeq2 (version 1.20.0) [60].

### 4.5. Validation of Transcriptome Data

To confirm the accuracy of our transcriptome analysis, RNA samples were reverse-transcribed into cDNA, which served as the template for quantitative real-time PCR (qRT-PCR). Gene-specific primers for DEGs were meticulously designed, with sequences detailed in Appendix A. The expression levels of DEGs were quantified by qRT-PCR using SYBR Green Fast qPCR Mix (ABclonal, Wuhan, China) on a QuantStudioTM 6 Flex System (Applied Biosystems, Waltham, MA, USA), adhering strictly to the manufacturer’s instructions.

### 4.6. Functional Annotation and Gene Ontology and Kyoto Encyclopedia of Genes and Genomes Classification

DEGs underwent enrichment analysis for GO functions and KEGG pathways using the cluster Profiler tool. This analysis encompassed GO categories such as molecular function (MF), cellular components (CCs), and biological processes (BPs). Terms and pathways with an adjusted *p*-value (padj) of less than 0.05 were deemed significantly enriched, highlighting the biological significance of the DEGs within these functional contexts.

### 4.7. Histochemical GUS Staining

Histochemical GUS assays were conducted using a 1% solution of 5-bromo-4-chloro-3-indolyl β-D-glucuronide (X-Gluc) in a 20 mM sodium phosphate buffer (pH 7.2), supplemented with 0.1% Triton X-100, 10 mM EDTA, and 5 mM potassium ferricyanide. *Arabidopsis* seedlings harboring the *ProHY5: GUS* construct were grown on agar plates with either 0.1 or 100 μM B for 7 days. The seedlings were then stained at 37 °C for 4 h. Post-staining, tissues were decolorized using 75% ethanol, and GUS staining patterns were documented under an optical microscope.

### 4.8. Determination of Nitrogen in Plants

Tissue samples were initially oven-dried at 105 °C for 30 min and subsequently at 65 °C until a constant weight as achieved. The dried samples were then weighed, finely ground, and soaked overnight in sulfuric acid. The mixture was boiled, and hydrogen peroxide was added dropwise to expedite digestion until the solution became clear and transparent. After diluting to volume with ultrapure water and filtering, the nitrogen concentration was quantified using a flow injection analyzer (Germany, Seal, AA3). 

### 4.9. Enzyme Activity Assay

The enzyme activity associated with nitrogen metabolism in the roots of *B. napus* was determined using an enzyme activity assay kit sourced from Boxbio (Boxbio, Beijing, China). The assay was conducted in strict accordance with the manufacturer’s protocol to ensure accurate measurement. measurement method provided by the manufacturer.

### 4.10. Statistical Analysis 

Each experiment was conducted using a completely randomized design with independent experiments replicated at least three times. Statistical analysis was executed utilizing SPSS16.0 for Windows. Significant differences were assessed using analysis of variance and Duncan’s test at the *p* < 0.05 level.

## Figures and Tables

**Figure 1 ijms-25-08319-f001:**
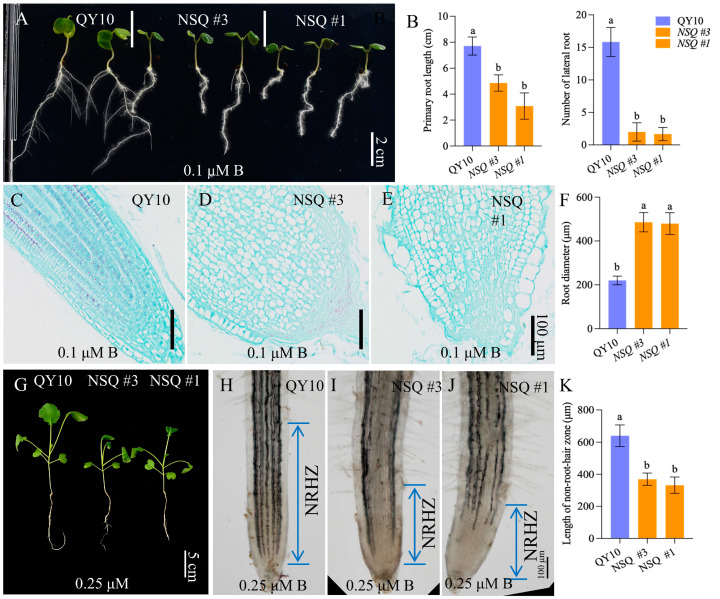
Downregulated expression of *BnaA3NIP5;1* enhances boron sensitivity. (**A**) The phenotypic characteristics and statistical analysis were conducted on primary root length and lateral root number in QY10 and NSQ lines cultured on plates for 10 days under 0.1 μM B. The study involved a minimum of three replicates. (**B**) Additionally, longitudinal sections (**C**–**E**) and diameter statistics (**F**) of the root tips from QY10 and NSQ lines in scenario (**A**) were examined (*n* ≥ 3). (**G**) Furthermore, the phenotypes of QY10 and NSQ lines were observed after 14 days of culture in nutrient solution under 0.25 μM B. (**H**–**J**) Microstructural analysis of the root tips from QY10 and NSQ lines (in scenario (**G**)) revealed the non-root-hair zone (NRHZ) length. (**K**) The NRHZ length was statistically analyzed (*n* ≥ 3), and the reported values represent means ± standard deviation. Letters denote significant differences between different treatments based on Duncan’s test (*p* < 0.05). The abbreviation ‘NSQ’ refers to the *BnaA3NIP5;1* RNAi lines.

**Figure 2 ijms-25-08319-f002:**
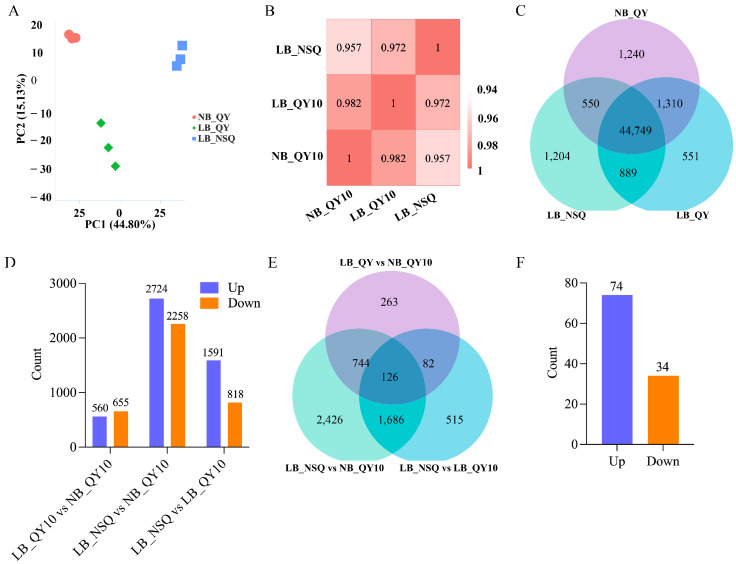
Differentially expressed genes (DEGs) among different groups. (**A**–**C**) We conducted principal component analysis (PCA), sample correlation analysis, and generated a gene coexpression Venn diagram using data from nine samples. These samples included NB_QY10 (normal boron QY10, 100 μM B), LB_QY110 (low-boron QY10, 0.25 μM B), and LB-NSQ (low-boron NSQ, 0.25 μM B), each with three biological replicates. (**D**) We quantified the differentially expressed genes (DEGs) between various groups. (**E**) Additionally, we created a Venn diagram to compare DEGs among LB_QY10 versus NB_QY10, LB_NSQ versus NB_QY10, and LB_NSQ versus LB_QY10. (**F**) Finally, we determined the count of upregulated and downregulated DEGs in the union of the DEGs identified in scenario (**E**).

**Figure 3 ijms-25-08319-f003:**
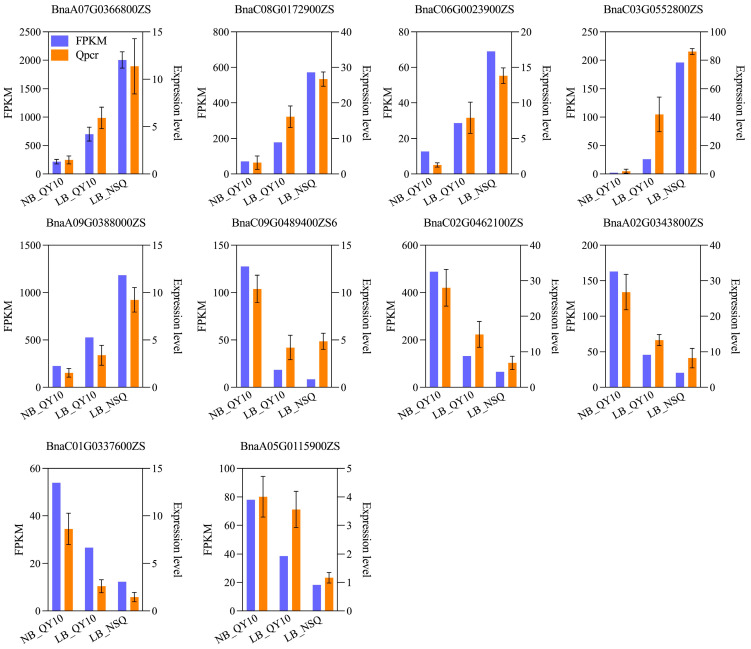
qRT-PCR verification transcriptome data results of selected 10 DEGs. The gene expression profiles were assessed using RNA-seq (**left panel**) and qRT-PCR (**right panel**) in NB_QY10, LB_QY10, and LB_NSQ. The study involved three pools, each containing 30–50 root tips, and the reported values represent means ± SD.

**Figure 4 ijms-25-08319-f004:**
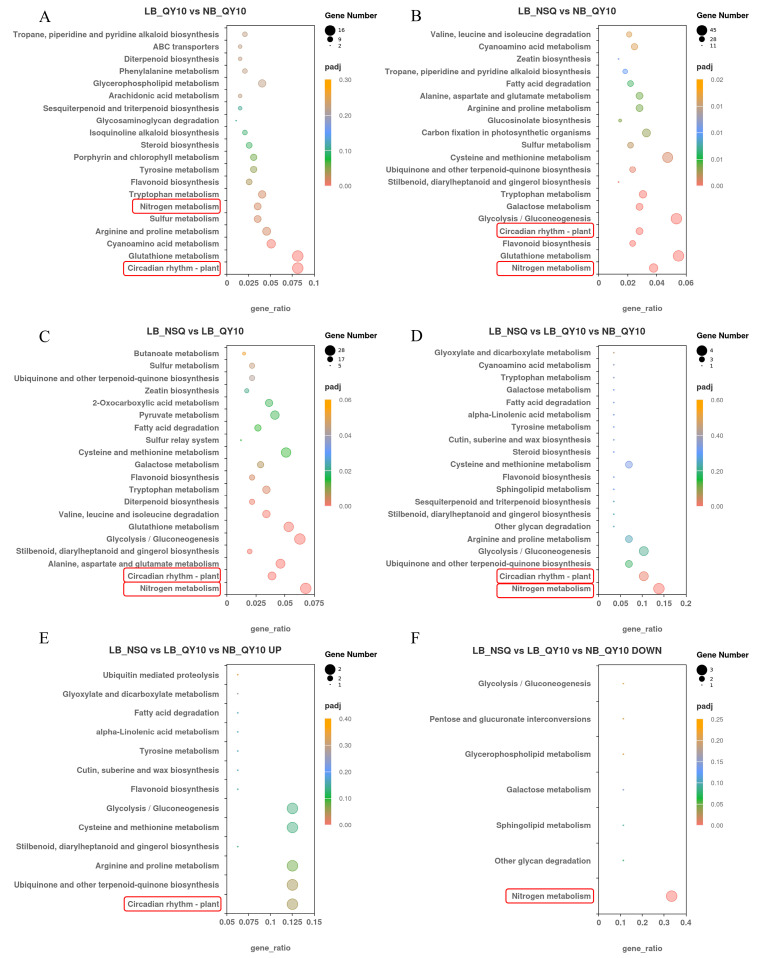
KEGG functional enrichment analysis of DEGs in different groups. (**A**) KEGG enrichment analysis of DEGs in LB_QY10 vs. NB_QY10. (**B**) KEGG enrichment analysis of DEGs in LB_NSQ vs. NB_QY10. (**C**) KEGG enrichment analysis of DEGs in LB_NSQ vs. LB_QY10. (**D**) KEGG enrichment analysis of DEGs in LB_NSQ vs. LB_QY10 vs NB_QY10. (**E**) KEGG enrichment analysis of up-regulated DEGs in LB_NSQ vs. LB_QY10 vs NB_QY10. (**F**) KEGG enrichment analysis of down-regulated DEGs in LB_NSQ vs. LB_QY10 vs NB_QY10. The dot size and color correspond to the number of genes and the corrected *p*-value, respectively. The gene ratio represents the proportion of differentially expressed genes (DEGs) annotated within a specific pathway term relative to the total number of genes annotated in that term. A higher gene ratio signifies greater pathway intensity. The padj value, ranging from 0 to 1, reflects the corrected *p*-value, with lower values indicating stronger significance. Only the top 20 enriched pathways are displayed, and the red box highlights the pathways of particular interest.

**Figure 5 ijms-25-08319-f005:**
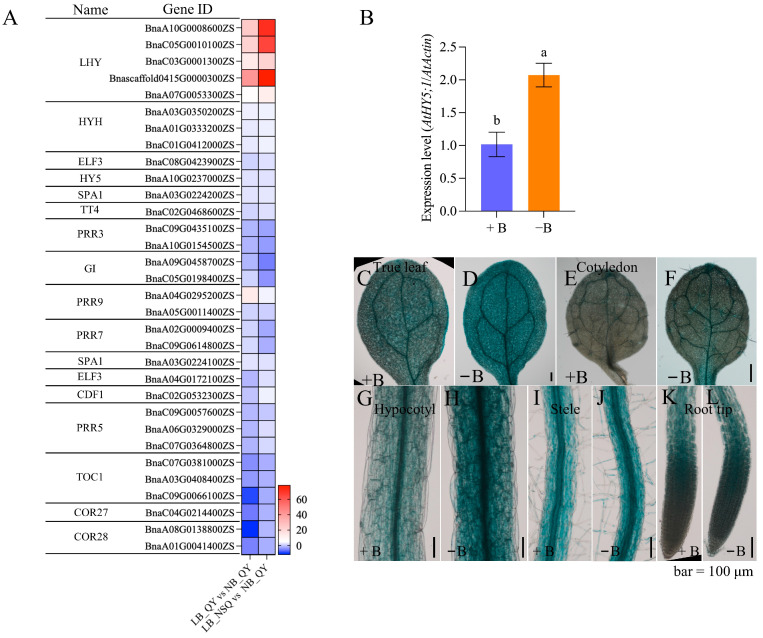
Expression patterns of circadian rhythm pathway genes in *B. napus* and *Arabidopsis*. (**A**) The gene identity (ID), gene annotation (name), and differential expression multiples of circadian rhythm pathway genes were investigated under two conditions: LB_QY10 versus NB_QY10 and LB_NSQ versus NB_QY10 using RNA-seq. (**B**) The expression pattern of *AtHY5* in *Arabidopsis* Col-0 roots was examined after 12 days of growth on agar plates under normal (100 μM B) and low-boron (0.1 μM B) conditions. The study involved three pools, each containing 20 roots, (*n* = 3). and the reported values represent means ± standard deviation. Letters denote significant differences between different treatments based on Duncan’s test (*p* < 0.05). Additionally, GUS staining was performed on 7-day-old *ProAtHY5: GUS* seedlings cultured on agar plates with normal boron ((**C**,**E**,**G**,**I**,**K**) (100 μM B)) and boron-deficient ((**D**,**F**,**H**,**J**,**L**) (0.1 μM B)) conditions.

**Figure 6 ijms-25-08319-f006:**
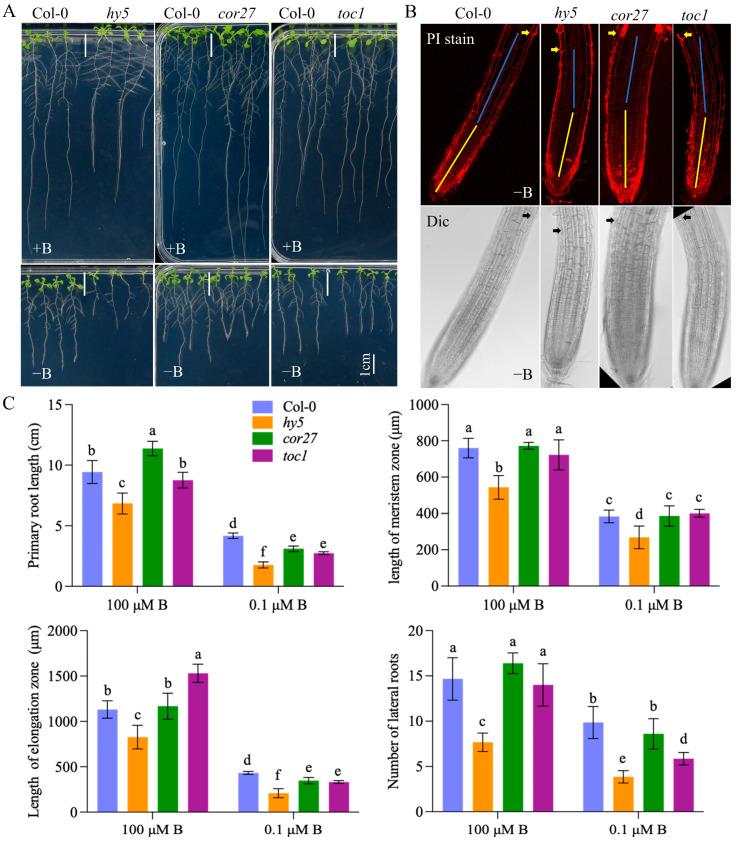
*hy5/cor27/toc1* mutants are more sensitive to boron deficiency than Col-0. (**A**) The phenotypes of wild-type (Col-0) and *hy5/cor27/toc1* mutant plants were observed after 12 days of growth on agar plates under both normal (100 μM B) and boron-deficient (0.1 μM B) conditions. (**B**) Microstructure images of the root tips from Col-0 and *hy5/cor27/toc1* mutants were captured under boron deficiency (0.1 μM B) conditions in scenario (**A**). The root tips were stained with propidium iodide (PI), and the upper row displays the view after PI staining, while the lower row shows the view under bright field. Arrows indicate the location of root hairs. (**C**) Statistical analysis was performed on primary root length, meristematic zone length, elongation zone length, and the number of lateral roots in Col-0 and *hy5/cor27/toc1* mutants from scenario (**A**) (*n* ≥ 10). The reported values represent means ± standard deviation. Letters denote significant differences between different treatments based on Duncan’s test (*p* < 0.05).

**Figure 7 ijms-25-08319-f007:**
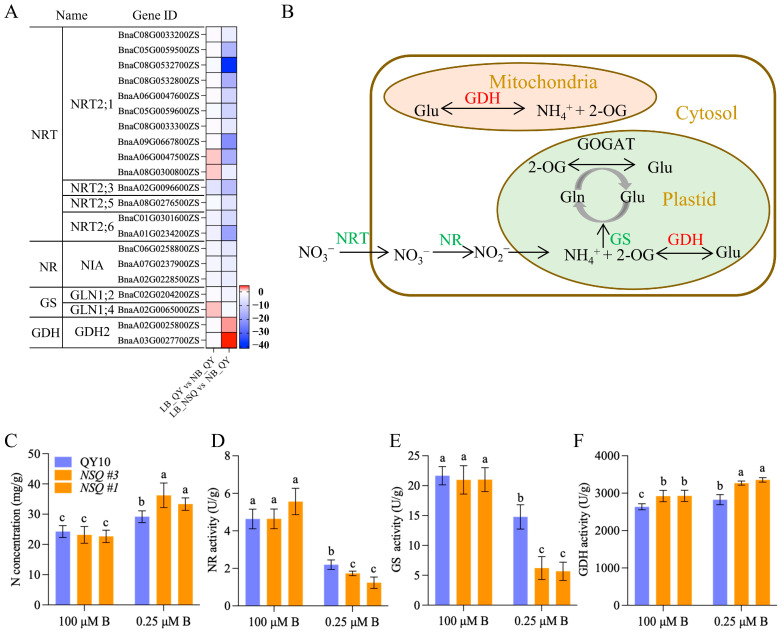
Nitrogen metabolism participates in the response of *B. napus* roots to B deficiency. (**A**) The gene identity (ID), gene annotation (name), and differential expression multiples of nitrogen metabolism pathway genes were analyzed in RNA-seq data for LB_QY10 versus NB_QY10 and LB_NSQ versus NB_QY10. (**B**) A schematic diagram illustrates transcriptome changes in the main pathways of nitrate absorption and assimilation. Genes in green font indicate decreased expression during the low-boron response, while genes in red represent increased expression. (**C**–**F**) Nitrogen concentration, nitrate reductase (NR), glutamine synthase (GS), and glutamate dehydrogenase (GDH) were determined in roots of Y10 and NSQ lines cultured in nutrient solution for 14 days under 100 and 0.25 μM B. The values represent means ± SD (standard deviation), and letters indicate significant differences between different treatments based on Duncan’s test (*p* < 0.05).

**Figure 8 ijms-25-08319-f008:**
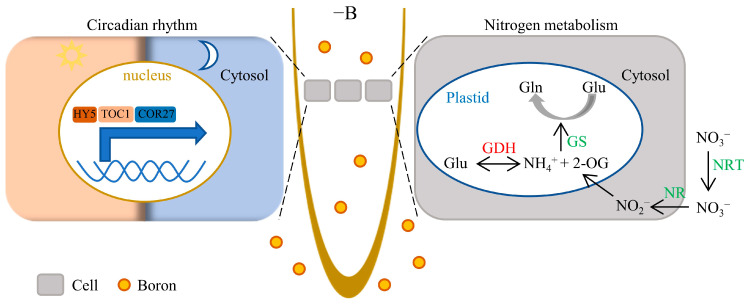
A working model illustrating the involvement of circadian rhythms and nitrogen metabolism in B deficiency-mediated root tip growth inhibition. Boron (B) deficiency activates the activity of circadian-rhythm-related transcription factors (**on the left**) and reduces the transcription level and enzyme activity of nitrogen metabolism-related genes (**on the right**, where green font represents decrease and red font represents increase). The gray box indicates cells, and the orange circle represents B.

## Data Availability

Data is contained within the article and Appendix A.

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
