# Peer review of "Circadian Rhythm and Nitrogen Metabolism Participate in the Response of Boron Deficiency in the Root of Brassica napus"

_ijms, 2024, doi:10.3390/ijms25158319_

Round 1
Reviewer 1 Report
Comments and Suggestions for Authors
General comments
The work is excellent, with a good beginning, middle, and end flow. The study showed how important B dynamics are in root tip cells, affecting circadian rhythm and nitrogen metabolism and ultimately slowing down growth in Brassica napus. This gave us useful information about how B regulation affects root tip development and overall root growth. In more detail, the researchers showed that BnaA3.NIP5;1 downregulation impacts root growth, circadian rhythm, and nitrogen metabolism. They also showed that Brassica napus root growth and nitrogen metabolism are hampered by a lack of B, and circadian rhythm genes like HY5 show similar B regulation patterns.
Specific comments
1. Novelty: The central question should be clearly defined at the end of the introduction section. To advance current knowledge, it is important to highlight novelty. The authors should revise the text from lines 81 to 87 in the introduction to present the novelty and hypothesis, and then proceed to discuss the objectives.
2. Scope: The work fits the IJMS.
3. Significance: The interpretation of the results was appropriate and significant, thereby justifying and supporting the conclusions.
4. Quality: The presentation of the data and analyses was appropriate. The figures were well-crafted and of excellent quality. However, there is a need to improve the quality of Figure 2 (ABC) and Figure 4, as they practically lack eligibility.
5. Scientific Soundness: The methodological approach is robust and well-described, indicating a well-conducted study.
6. Statistical analysis: We must evaluate the data for normality (Shapiro-Wilk test, for example) and homogeneity of variances (Levene test, for example).
7. Interest to the Readers: Readers may be interested in the following contributions: 1. RNA-seq technology was used to study how B deficiency affects root development, and 2. physiological pathways and indicators for root development insights were found and put together. As a result, these aspects may pique readers' interest in the field of knowledge.
8. Overall Merit: Despite the absence of references, the work is generally well-described, with an adequate methodology and appropriate discussion.
9. English Level: The text has a clear flow and is appropriate and understandable. However, before possible publication, the editorial staff should make a careful revision.
A crucial point that deserves attention
The biggest obstacle was the lack of a list of references. This made reading and interpreting the discussion very complicated, despite the quality of the manuscript. Thus, the authors should resubmit the manuscript, including references.
Comments on the Quality of English Languagegood
Author Response
Response to reviewer 1
- Novelty: The central question should be clearly defined at the end of the introduction section. To advance current knowledge, it is important to highlight novelty. The authors should revise the text from lines 81 to 87 in the introduction to present the novelty and hypothesis, and then proceed to discuss the objectives
Response:Thank you for your suggestion. We have revised the format and other aspects of the article.
- Scope: The work fits the IJMS.
Response:Thank you for your comment.
- Significance: The interpretation of the results was appropriate and significant, thereby justifying and supporting the conclusions.
Response:Thank you for your comment.
- Quality: The presentation of the data and analyses was appropriate. The figures were well-crafted and of excellent quality. However, there is a need to improve the quality of Figure 2 (ABC) and Figure 4, as they practically lack eligibility.
Response:Thank you for your suggestion. We have revised Figures 2 and 4 and presented them in the revised manuscript.
- Scientific Soundness: The methodological approach is robust and well-described, indicating a well-conducted study.
Response:Thank you for recognizing our research work.
- Statistical analysis: We must evaluate the data for normality (Shapiro-Wilk test, for example) and homogeneity of variances (Levene test, for example).
Response:Thanks for your suggestion. The general analysis methods for transcriptome analysis are described as follows: The gene expression values of RNA seq are generally calibrated using FPKM, which corrects for sequencing depth and gene length (Bray et al., 2015). After calculating the expression values of all genes in each sample (FPKM), we used a box plot to display the distribution of gene expression levels in different samples, as shown in the following figure (Figure 1). To analyze the differences in gene expression levels between groups, we first used the DESeq standardization method in the DESeq2 software to standardize the data. Then, we used a negative binomial distribution model to calculate the probability of hypothesis testing (Pvalue), and finally performed multiple hypothesis testing correction to obtain the FDR value (false discovery rate, padj is a common form) (Anders et al., 2010; Robinson et al., 2010; Love et al., 2014). And |log2 (FoldChange)| ≥ 1 & padj ≤ 0.05 were used as screening criteria to screen for differentially expressed genes.
Figure 1. The distribution of expression levels of all genes in each sample. The horizontal axis in the figure represents the sample name, and the vertical axis is log2 (FPKM+1). The box plots for each region correspond to five statistical measures (maximum, upper quartile, median, lower quartile, and minimum).
Bray N, Pimentel H, Melsted P, et al. Near-optimal RNA-Seq quantification. arXiv preprint arXiv:1505.02710, 2015.
Anders S, Huber W. Differential expression analysis for sequence count data. Genome biol, 2010, 11(10): R106.
Love M I, Huber W, Anders S. Moderated estimation of fold change and dispersion for RNA-seq data with DESeq2. Genome biology, 2014, 15(12): 1−21. (DESeq2)
Robinson M D, McCarthy D J, Smyth G K. edgeR: a Bioconductor package for differential expression analysis of digital gene expression data. Bioinformatics, 2010, 26(1): 139−140. (edgeR)
- Interest to the Readers: Readers may be interested in the following contributions: 1. RNA-seq technology was used to study how B deficiency affects root development, and 2. physiological pathways and indicators for root development insights were found and put together. As a result, these aspects may pique readers' interest in the field of knowledge.
Response:Thank you very much for your review. In our work, we first evaluated the importance of boron supply from the lateral root cap for root development under low boron stress. Subsequently, we evaluated the physiological pathways using RNA-seq tool, and validated these through genetic analysis and physiological measurements. This provides a basis for identifying important genes regulating root development under low boron stress.
- Overall Merit: Despite the absence of references, the work is generally well-described, with an adequate methodology and appropriate discussion
Response:Thank you very much for your positive evaluation. We have included references which were missed in former by a misoperation in revised manuscript.
- English Level: The text has a clear flow and is appropriate and understandable. However, before possible publication, the editorial staff should make a careful revision.
Response:Thank you for your suggestion. We have revised the format and other aspects of the article.
- 10.A crucial point that deserves attention. The biggest obstacle was the lack of a list of references. This made reading and interpreting the discussion very complicated, despite the quality of the manuscript. Thus, the authors should resubmit the manuscript, including references.
Response:Firstly, I am deeply sorry for our carelessness in forgetting to attach references, which has increased the difficulty of your reading. Secondly, thank you for your recognition of our manuscript. We have added the references to the newly revised manuscript.

Reviewer 2 Report
Comments and Suggestions for Authors
This study was focused on circadian rhythm and nitrogen metabolism in the response of boron deficiency in the root of Brassica napus. The Authors investigated the role of BnaA3.NIP5;1, a gene encoding a boric acid channel in B. napus. Downregulation of BnaA3.NIP5;1 enhanced B sensitivity in B. napus, resulting in reduced shoot biomass and impaired root tip development. Differentially expressed genes (DEGs) were significantly enriched in plant circadian rhythm and nitrogen (N) metabolism pathways. The circadian rhythm-related gene HY5 exhibited a similar B regulation pattern in Arabidopsis as observed in B. napus. Furthermore, Arabidopsis mutants with disrupted circadian rhythm (hy5/cor27/toc1) displayed heightened sensitivity to low B compared to the wild type (Col-0). Boron deficiency significantly disrupted N metabolism in B. napus roots, affecting nitrogen concentration, nitrate reductase enzyme activity, and glutamine synthesis. This disruption was exacerbated in BnaA3NIP5;1 RNAi lines.
The paper is well organized and important in the described research field. However, I have formulated some significant improvements listed below:
- Authors did not include References (there are no list of the citations at the end of the manuscript). Therefore, it has to be added.
- I recommend including the electropherograms presenting the RNA bands in agarose gels in the Supplementary file – it would provide information regarding quality of total RNA samples. In addition, RIN (RNA integrity number) should be included in the manuscript or in the Supplementary file.
- Authors used SYBR Green fluorescent dye during RT-PCR gene expression studies, hence, the results of Melting Curve Analysis should be obligatory added in the manuscript or Supplementary file (e.g., JPG or TIFF file).
- Graphical resolution of the figure 4 should be considerably increased.
- Moderate editing of English language required.
Comments on the Quality of English LanguageModerate editing of English language required.
Author Response
Response to reviewer 2
- This study was focused on circadian rhythm and nitrogen metabolism in the response of boron deficiency in the root of Brassica napus. The Authors investigated the role of BnaA3.NIP5;1, a gene encoding a boric acid channel in napus. Downregulation of BnaA3.NIP5;1 enhanced B sensitivity in B. napus, resulting in reduced shoot biomass and impaired root tip development. Differentially expressed genes (DEGs) were significantly enriched in plant circadian rhythm and nitrogen (N) metabolism pathways. The circadian rhythm-related gene HY5 exhibited a similar B regulation pattern in Arabidopsisas observed in B. napus. Furthermore, Arabidopsis mutants with disrupted circadian rhythm (hy5/cor27/toc1) displayed heightened sensitivity to low B compared to the wild type (Col-0). Boron deficiency significantly disrupted N metabolism in B. napus roots, affecting nitrogen concentration, nitrate reductase enzyme activity, and glutamine synthesis. This disruption was exacerbated in BnaA3NIP5;1 RNAi lines.
Response: We highly appreciate your positive comments and invaluable suggestions. We have revised the manuscript accordingly and answered the questions as follows.
The paper is well organized and important in the described research field. However, I have formulated some significant improvements listed below:
- Authors did not include References (there are no list of the citations at the end of the manuscript). Therefore, it has to be added.
Response:Thank you very much for your suggestion. We have added references in the revised manuscript.
- I recommend including the electropherograms presenting the RNA bands in agarose gels in the Supplementary file – it would provide information regarding quality of total RNA samples. In addition, RIN (RNA integrity number) should be included in the manuscript or in the Supplementary file.
Response:Thank you very much for your suggestion. The RNA quality testing was carried out by the Fragment Analyzer system (Agilent 5400). Based on this system, the RNA concentration and integrity of all samples were reflected in Supplementary Figures S3−S5 and Table S1 in the revised manuscript.
- Authors used SYBR Green fluorescent dye during RT-PCR gene expression studies, hence, the results of Melting Curve Analysis should be obligatory added in the manuscript or Supplementary file (e.g., JPG or TIFF file).
Response:Thank you for your suggestion. We have added the results of Melting Curve Analysis in the revised manuscript (Supplement Figure S7).
- Graphical resolution of the Figure 4 should be considerably increased.
Response:Thank you very much for your suggestion. We have plotted Figure 4 to improve its clarity and meet the publication requirements. We have included the new image in the revised manuscript.
- Moderate editing of English language required.
Response:Thank you for your suggestion. We have made some language modifications in the revised manuscript.

Round 2
Reviewer 1 Report
Comments and Suggestions for Authors
The manuscript has been significantly improved compared to the first submitted version. The authors have adequately addressed my questions. Therefore, the manuscript can be accepted for publication.
Reviewer 2 Report
Comments and Suggestions for Authors
The manuscript has been sufficiently improved. I recommend to accept it in present form.
Comments on the Quality of English LanguageMinor editing of English language required